# Social determinants of health in lesbian, gay, bisexual, transgender, queer, and other sexual and gender minority (LGBTQ+) older adults: Impact of socioeconomic disadvantage on inpatient hospitalizations

Jennifer T. May[1]*, Devon Noonan[2], Susan G. Silva[2]

**1** Department of Biobehavioral Health & Nursing Science, College of Nursing, University of South Carolina, Columbia, South Carolina, United States of America, **2** School of Nursing, Duke University, Durham, North Carolina, United States of America

* jm293@mailbox.sc.edu

## Abstract

### Introduction

Little is known about the impact of socioeconomic disadvantage on lesbian, gay, bisexual, transgender, queer, and other sexual and gender minority (LGBTQ+) older adults (≥50 years). The aim of this study is to determine whether the distribution of LGBTQ+ inpatient hospitalizations are related to structural socioeconomic factors.

### Methods

A secondary analysis of retrospective electronic health record data for LGBTQ+ older adults hospitalized from 2018 to 2022 was conducted at one large health system. The average county area deprivation index where the patient resided was calculated.

### Results

The analysis included 2270 LGBTQ+ older adult inpatient hospitalizations, with 1508 (66.4%) from low socioeconomic disadvantage, 595 (26.3%) from moderate socioeconomic disadvantage; and 17 (7.4%) from high socioeconomic disadvantage counties (p < .0001). LGBTQ+ older adults who resided in moderate and high socioeconomic disadvantaged counties had a significant proportion of patients identifying as asexual (*a posteriori* contrasts, p < .05) compared to the low socioeconomic disadvantaged group. Those from moderate socioeconomic disadvantaged counties had a significantly higher proportion of patients identifying as bisexual (*a posteriori* contrasts, p < .05) compared to the high socioeconomic disadvantaged group.

**Data availability statement:** The data used in this study electronic health record data from the Duke University Health System. Data may be available upon request to the Duke Office of Clinical Research at docr.help@dm.duke.edu or the Duke IRB at https://irb.duhs.duke.edu/about-us/contact-us or call 919.668.5111. The data is not readily available because it is restricted due to HIPPA protections and contains sensitive information. Additionally, the data has to be extracted by an honest broker with Duke Health System and then placed into a protected environment for data analysis. Only data tables (those in the paper) can be downloaded and have to be approved by the honest broker before they can be used.

**Funding:** This work was supported by the National Center for Advancing Translational Sciences (NCATS), National Institutes of Health, through Grant Award Number UL1 TR002553. The content is solely the responsibility of the authors and does not necessarily represent the official views of the NIH.

**Competing interests:** The authors have declared that no competing interests exist.

## Discussion

This analysis highlights socioeconomic disadvantage of LGBTQ+ older adults who utilized one large health system. More work needs to be done to understand use of the hospital system by LGBTQ+ older adults in moderate to high socioeconomic disadvantaged areas.

## Introduction

Lesbian, gay, bisexual, transgender, queer, and other sexual and gender minority (LGBTQ+) older adults experience significant health disparities compared to their heterosexual and cisgender counterparts. These disparities are evident across various health outcomes, including higher rates of chronic conditions, mental health issues, and barriers to accessing healthcare services [1,2]. LGBTQ+ older adults face distinct challenges, such as being twice as likely to live alone, less likely to have children, and more vulnerable to poverty, homelessness, and both physical and mental health issues [1]. In North Carolina, where this study is located, 30% of LGBTQ+ adults aged 18+have an annual income of below $24,000, 21% are uninsured (compared to 14% for non-LGBTQ+ adults), and 29% are food insecure (compared to 16% for non-LGBTQ+ adults) [3].

The health disparities experienced by LGBTQ+ older adults are influenced by both individual factors (such as sexual orientation, gender identity, and age) and social determinants of health (SDOH). Structural and social determinants, including racism, socioeconomic status, education level, employment, housing quality, and environment, contriubute to health inequities [4,5]. For LGBTQ+ older adults, these factors intersect with their sexual orientation and gender identity, potentially exacerbating health risks. Moving beyond individual-level predictors to understand how social structures and environments influence SDOH in marginalized populations can provide insight into expanding knowledge on LGBTQ+ health inequities.

Residential neighborhood and county-level characteristics have been shown to be important predictors of health outcomes [6,7]. Residing in socially disadvantaged areas (areas that suffer from a combination of economic, health, and environmental burdens, including poverty, high unemployment, pollution, and high incidence of chronic disease) is associated with increased health risks, including higher rates of heart failure, hospitalizations, and mortality [4,8,9]. Previous studies have demonstrated that living in a socioeconomically disadvantaged area predicts higher rates of hospitalization, rehospitalization, and emergency surgeries [10–12]. These findings are particularly concerning for LGBTQ+ older adults, who historically have lower incomes and higher poverty levels compared to non-LGBTQ+ populations. Furthermore, LGBTQ+ individuals face significant challenges related to homelessness, with 3% of sexual minorities and 8% of transgender adults having experienced homelessness in the last 12 months [13]. While these socioeconomic challenges undoubtedly influence access to healthcare, it remains unclear whether LGBTQ+ older adults face additional barriers when living in more socioeconomically disadvantaged areas.

Recent studies have indicated that county-level characteristics (sociodemographic, health-related, environmental) are related to higher rates of COVID-19 deaths [14], malnutrition in older adults [15], and individual tobacco and alcohol use [16]. Those living in higher socioeconomically disadvantaged areas are less likely to have controlled blood pressure, diabetes, and cholesterol compared to those living in lower socioeconomic disadvantaged areas [17]. However, little is known about the impact of area-level socioeconomic disadvantage on LGBTQ+ older adult inpatient hospitalizations and health outcomes.

Despite the growing body of evidence on health disparities among LGBTQ+ older adults and the influence of community-level factors on health outcomes, there is a significant gap in understanding how county-level deprivation specifically affects the health of LGBTQ+ older adults (≥50 years). This study aimed to examine the relationship between county-level socioeconomic disadvantage and health outcomes in a North Carolina hospital system. Understanding this relationship can provide crucial information to health systems serving this population and guide the development of targeted interventions, policies, and resource allocation to advance health equity for LGBTQ+ older adults.

## Sexual and gender minority health research framework

Adapted from the Minority Health and Health Disparities Research Framework, the Sexual and Gender Minority (SGM) Health Research Framework [18] (Fig 1) is a multilevel model that shows the key factors and influences of LGBTQ+ health throughout the life span. Guided by the social ecological model [19], the SGM Health Research Framework has four key levels that influence the health and well-being of LGBTQ+ populations: individual factors, interpersonal factors, community factors, and societal factors. In the model each level is encapsulated by the next, indicating the intersectionality of each factor. At the individual level personal, behavioral, biological, and demographics drive health and well-being [19]. The interpersonal level focuses on relationships and other social networks and how that influences the health of the individual [18]. Community factors include the places where LGBTQ+ populations have social interactions. This includes places like neighborhoods, health care settings, and community centers [18]. Societal factors, the last factor in this framework, consider influences like policy (all levels), sexual orientation/gender identity data collection practices, and legal protections as factors LGBTQ+ populations have to navigate [18].

## Area Deprivation Index (ADI)

As discussed above in relation to the SGM Health Research Framework, community factors influence LGBTQ+ older adult health throughout the lifespan. ADI is a community level measure that is comprised of 17 components over 4 categories (see Table 1) from the 5-year American Community Survey. ADI provides an index score of socioeconomic disadvantage based on the census block groups of the neighborhood [20]. The importance of the ADI is that it provides a comprehensive understanding of the social and economic conditions within a given region and can be used to provide information to target interventions and resources for areas with the greatest need.

## Aims

This study focuses on LGBTQ+ older adults with at least one inpatient hospitalization in a large academic health system in North Carolina to determine whether the distribution of the inpatient hospitalizations is related to the average ADI for the county in which the patient resides. The academic health system is world-renowned and centrally located in North Carolina. It is a comprehensive care center and includes a regional emergency and trauma center. We examined inpatient hospitalizations as a key indicator of healthcare utilization and potential health disparities across areas with varying levels of socioeconomic deprivation. We hypothesized that counties with a higher county average ADI, indicative of greater area-level socioeconomic disadvantage, would have a significantly higher proportion of the LGBTQ+ older adult inpatient hospitalizations. We use the SGM Health Disparities Research Framework [18] in this study to describe our results, identify gaps, and to guide future research and interventions within the levels of influence (individual, interpersonal, community).

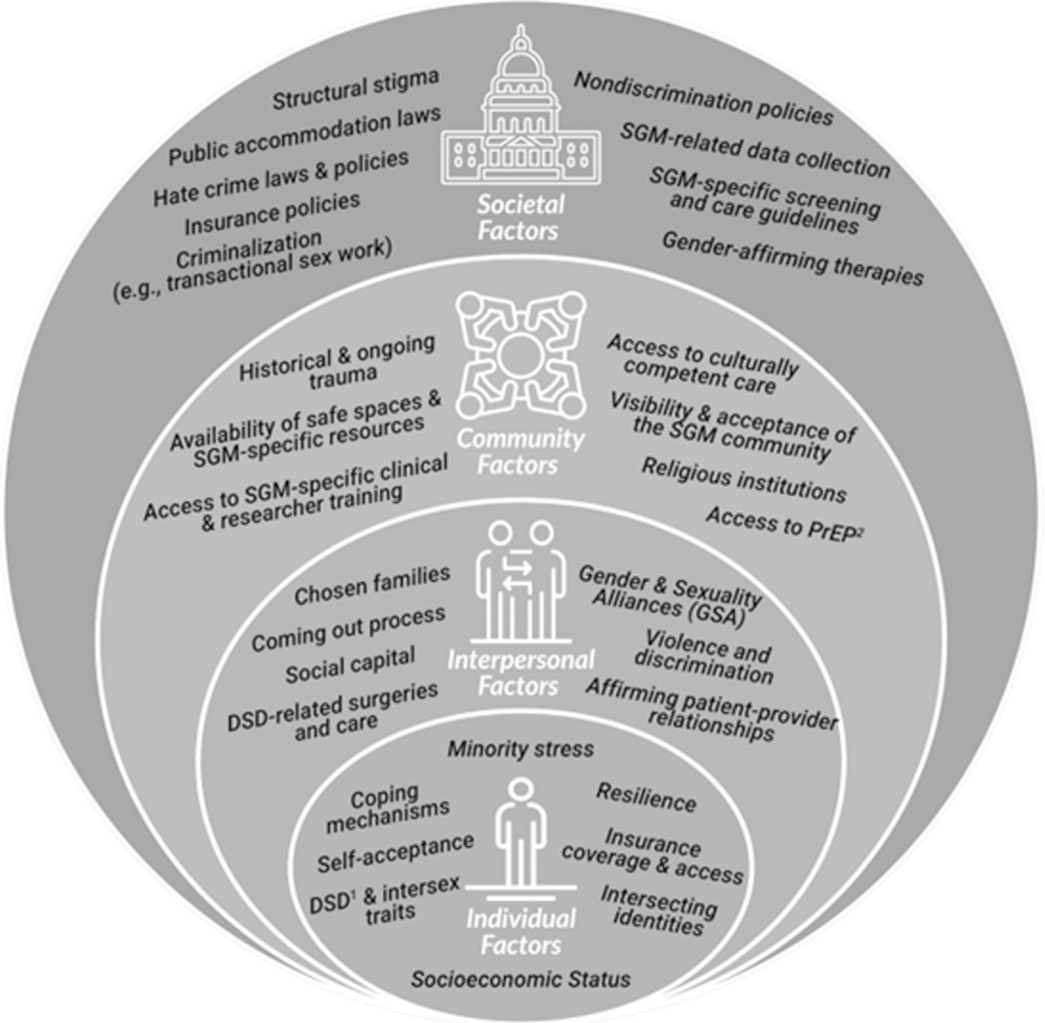

**Fig 1. Sexual & gender minority health disparities research framework.**

## Methods

### Study design and participants

This descriptive study was a secondary analysis of retrospective electronic health record (EHR) data from a large academic health system. We included inpatient hospitalizations for adult patients who identified as LGBTQ+ who were ≥50 years old with a primary diagnosis and were inpatient admissions to the health system one or more times from November 1, 2018 to June 1, 2022. Only primary diagnoses were available for analysis. We excluded emergency room visits. There were a total of 2284 hospitalizations; 14 hospitalizations did not have county level data. Therefore, a total of 2270 LGBTQ+ older adult inpatient hospitalizations were studied, representing 1235 unique patients. An honest broker employed by the University was used to retrieve the data based on our inclusion and exclusion criteria and uploaded it in the healthsystem's secure environment be be analyzed. The data was de-identified and consent was not required for this secondary data analysis. This study was approved by Duke University Institutional Review Board (Pro00110074).

**Table 1. Area deprivation index indicators.**

| Domain | Variables |
|---|---|
| Education | % Population aged 25+ with less than 9 years of education<br>% Population aged 25+ with at least a high school diploma<br>% Population aged 16+ employed in white-collar occupations |
| Employment/Income | Income disparity<br>Median family income<br>% Civilian labor force unemployed (aged 16+)<br>% Families below poverty level<br>% Population below 150% of poverty threshold |
| Housing | Median home value<br>Median monthly mortgage<br>Median gross rent<br>% Owner-occupied housing units<br>% Occupied housing units without complete plumbing |
| Household Characteristics | % Occupied housing units with more than 1 person per room<br>% Single-parent households with children under 18 years<br>% Households without a motor vehicle<br>% Households without a telephone |

## Measures

Individual, interpersonal, and community measures were guided by the SGM Health Disparities Research Framework. Individual measures for the LGBTQ+ older adults hospitalized were gender identity, sexual orientation, race, ethnicity, age, primary diagnosis, and number of hospitalizations per patient. Sexual orientation and gender identity (SOGI) fields in the EHR are listed in Table 2. The process of collecting SOGI data in this hospital system can be completed by the patient using their online chart or it can be entered in the clinical setting by the patient, licensed or nonlicensed staff who are in patient facing roles (see blinded for review [21]) for further explanation). The interpersonal measure was married/partnered. Neighborhood level data was not available for this study. A key community measure for each hospitalization was the area-level socioeconomic disadvantage for the patient at the time of the admission, defined as the average ADI for the county in which the patient resided. The average ADI for each county in North Carolina was calculated as follows: 2020 composite scores for the ADI for block groups in North Carolina were downloaded using the Neighborhood Atlas at the University of Wisconsin [22]. Census geocodes for the county neighborhoods were grouped by county using the assigned federal information processing standard (FIPS) code using the North Carolina State County FIPS table [23]. For each county, the average ADI was determined by averaging all the county census block geocodes [24]. County average ADI values ranged from 0 to 10, with a higher average ADI indicating greater area-level socioeconomic disadvantage. Each county was categorized into one of three socioeconomic disadvantage groups: (a) low – average ADI scores ranged from 0.00 to 3.99; (b) moderate – average ADI scores ranged from 4.00 to 6.99; and (c) high – average ADI scores ranged from 7.0 to 10.00. For each hospitalization, the area-level socioeconomic disadvantage category (ADI group) was then determined for the patient admitted. Societal factors were not measured in this study and therefore were not included in the analysis.

## Data analysis

The data analysis was conducted in a Protected Analytics Computing Environment (PACE) required by the healthsystem to ensure patient data privacy. Descriptive statistics were used to detail characteristics of the 1235 LGBTQ+ patients at the first inpatient hospitalization. Chi-square goodness of fit test was used to test for significant differences in the proportion of the 2270 LGBTQ+ older adults inpatient hospitalizations in which the patient resided in the low, moderate, or high socioeconomic disadvantage area (ADI groups). Chi-square/Fisher's Exact test for categorical characteristics and

**Table 2. EHR gender identity and sexual orientation fields.**

| Gender identity | Sexual orientation |
|---|---|
| Male | Asexual |
| Female | Bisexual |
| Nonbinary | Pansexual |
| Gender fluid/queer | Gay/Lesbian |
| Transgender female/male to female | Straight (not lesbian or gay) |
| Transgender male/female to male | Something else |
| Choose not to answer | Choose not to disclose |
|  | Do not know |

*Abbreviation*: EHR: electronic health record.

one-way analysis of variance for age were used to compare the three ADI groups with regard to patient characteristics at the time of the hospitalization. *A posteriori* pairwise contrast were performed when a significant overall effect was detected. Non-directional statistical tests were performed with significance set at 0.05 for each test. All statistical analysis were performed using SAS® software, version 9.4 [25].

## Results

### Patient characteristics

Table 3 presents the characteristics of the LGBTQ+ older adults at their first inpatient hospitalization (N = 1235 unique patients). The mean age was 66.8 years (SD = 9.9, range: 50–103). Most identified as male (52%), followed by female (47%), gender fluid (0.3%), transgender (0.2%), and non-binary (0.2%). Sexual orientation is reported as asexual (60%), lesbian or gay (27%), bisexual (11%), queer (0.6%), and pansexual (0.6%). The majority were Caucasian/White (81%) and married/partnered (58%). The top three primary diagnosis of the inpatient hospitalizations were all pulmonary related with nonspecific abnormal pulmonary finding of lung field (29.7%) being the majority. A total of 323 (26%) of the 1235 LGBTQ+ older adults had more than one inpatient hospitalization during the observation period.

### ADI groups and hospitalizations

There were a total of 2270 LGBTQ+ older adult inpatient hospitalizations, with 1508 (66.4%) inpatient hospitalizations in which the patient resided in a low socioeconomic disadvantage area, 595 (26.3%) inpatient hospitalizations in which the patient resided in a moderate socioeconomic disadvantage area, and 17 (7.4%) inpatient hospitalizations in which the patient resided in a high socioeconomic disadvantage area (Table 4, chi-square goodness of fit, p < .001). Table 4 details the county average ADI for each of the three ADI groups.

### ADI groups and patient characteristics

Table 5 presents the patient characteristics for the low, moderate and high socioeconomic disadvantage area groups. The three ADI groups significantly differed on several sexual orientation characteristics (overall effect, p < .05). The moderate and high groups had a significant proportion of patients reporting being asexual compared to the low group, while the low group had a higher proportion of patients reporting to be lesbian compared to the moderate and high groups (*a posteriori* contrasts, p < .05). The moderate group had a significantly higher proportion of patients reporting to be bisexual compared to the high group (*a posteriori* contrast, p < .05). In terms of demographics, the moderate group had a significantly higher

**Table 3. Patient characteristics at first hospitalization (N = 1235 patients).**

| Characteristic | n (%) |
|---|---|
| **Gender Identity** | |
| Male | 647 (52.4%) |
| Female | 579 (47.0%) |
| Gender Fluid | 4 (0.3%) |
| Transgender | 3 (0.2%) |
| Non-binary | 2 (0.2%) |
| **Sexual Orientation** | |
| Asexual | 740 (60.0%) |
| Lesbian or Gay | 339 (27.5%) |
| Bisexual | 142 (11.5%) |
| Pansexual | 7 (0.6%) |
| Queer | 7 (0.6%) |
| **Race** | |
| White | 1003 (81.2%) |
| Black or African American | 184 (15.0%) |
| Other Minorities | 41 (2.2%) |
| More than one race | 1 (0.1%) |
| Not Reported/Declined | 6 (0.5%) |
| **Ethnicity** | |
| Hispanic or Latinx | 24 (1.69%) |
| Non-Hispanic or Non-Latinx | 1168 (94.6%) |
| Not Reported/Declined | 43 (3.5%) |
| **Age, in years** | 66.8 ± 9.9 |
| **Top 3 Primary Diagnoses** | |
| Other nonspecific abnormal finding of lung field | 367 (29.7%) |
| Pleural effusion, not elsewhere classified | 176 (14.3%) |
| Shortness of breath | 143 (11.6%%) |
| **Hospitalizations per Patient** | |
| 1 hospitalization | 786 (63.6%) |
| 2 hospitalizations | 224 (18.1%) |
| 3 or more hospitalizations | 99 (8.0%) |
| **Married/partnered** | 723 (58.5%) |

proportion of patients who identified as White compared to the low and high groups as well as a higher proportion who were married/partnered compared to the low group (*a posteriori* contrasts, p < .05).

## Discussion

The primary finding of this study, which aimed to examine the relationship between socioeconomic disadvantage and inpatient hospitalization among LGBTQ+ older adults, was unexpected. Contrary to our initial hypothesis, the majority (66.4%) of hospitalized LGBTQ+ older adults in our sample resided in areas of low socioeconomic disadvantage. This finding challenges common assumptions about the relationship between LGBTQ+ status, aging, and socioeconomic status.

**Table 4. ADI groups: Hospitalizations and county average ADI (N = 2270 hospitalizations).**

| ADI Group Characteristics | County Average ADI: 0–3 Low Socioeconomic Disadvantage | County Average ADI: 4–6 Moderate Socioeconomic Disadvantage | County Average ADI: 7–10 High Socioeconomic Disadvantage | Chi-square Goodness of Fit Test, p-value |
|---|---|---|---|---|
| n (%) of 2270 hospitalizations | 1508 (66.4%) | 595 (26.2%) | 167 (7.4%) | <.001 |
| ADI: Mean ± SD | 3.3 ± 0.6 | 5.9 ± 0.7 | 7.8 ± 0.7 | |
| ADI: Minimum, maximum | 2.4, 3.9 | 4.1, 6.8 | 7.1, 9.4 | |

Note: Goodness of fit test: null hypothesis is the proportion for each ADI group will be equal (0.33 or 33%); ADI groups were categorized by county average ADI; SD = Standard Deviation.

**Table 5. ADI groups: Patient characteristics (N = 2270 hospitalizations)*.**

| Characteristic | County Average ADI: 0–3 Low Socioeconomic Disadvantage (L) N = 1508 | County Average ADI: 4–6 Moderate Socioeconomic Disadvantage (M) N = 595 | County Average ADI: 7–10 High Socioeconomic Disadvantage (H) N = 167 | Overall *p*-value | *A posteriori* Pairwise Contrasts |
|---|---|---|---|---|---|
| Gender identity | | | | | |
| Female | 699 (46.4%) | 273 (45.9%) | 89 (53.3%) | .21 | -- |
| Male | 798 (52.9%) | 322 (54.1%) | 78 (46.7%) | .23 | -- |
| Transgender | 5 (0.3%) | 0 (0.0%) | 0 (0.0%) | -- | -- |
| Gender Fluid | 4 (0.3%) | 0 (0.0%) | 0 (0.0%) | -- | -- |
| Non-binary | 2 (0.1%) | 0 (0.0%) | 0 (0.0%) | -- | -- |
| Sexual orientation | | | | | |
| Asexual | 874 (58.0%) | 416 (70.0%) | 121 (72.5%) | <.001 | (M = H) > L |
| Lesbian/Gay | 446 (30.0%) | 96 (16.1%) | 34 (20.4%) | <.001 | L > (M = H) |
| Bisexual | 156 (10.3%) | 79 (13.3%) | 12 (7.2%) | .05 | M > H |
| Queer | 21 (1.4%) | 0 (0.0%) | 0 (0.0%) | -- | -- |
| Pansexual | 11 (0.7%) | 4 (0.7%) | 0 (0.0%) | -- | -- |
| Age, in years | 67.5 ± 9.9 | 67.2 ± 9.4 | 68.2 ± 9.1 | .51 | -- |
| White race | 1164 (77.6%) | 526 (88.4%) | 126 (75.9%) | <.001 | M > (L = H) |
| Hispanic | 20 (1.4%) | 7 (1.2%) | 6 (3.7%) | .08 | -- |
| Married/partnered | 808 (53.6%) | 406 (68.4%) | 101 (60.8%) | <.001 | M > L |

Note: n (%) reported and 3 x 2 chi-square tests or Fisher Exact Tests with continuity correction for categorical characteristics. Mean ± standard deviation reported for age with one-way analysis of variance using a General Linear Model conducted due to unequal sample size. *A posteriori* pairwise contrasts with p ≤ 0.05 are indicated; '--' indicates test not conducted due to small number of cases in one or more ADI groups with the specified characteristic.

## Interpreting results through the SGM health disparities framework

The SGM Health Disparities Framework emphasizes the interplay of individual, interpersonal, community, and societal factors in shaping health outcomes for LGBTQ+ populations across the life course. Our results can be interpreted through this lens:

**Individual level factors.** The predominance of patients from low socioeconomic disadvantaged areas warrants further consideration of several contributing factors. This finding may reflect individual-level resilience and resource accumulation

over the life course, as LGBTQ+ older aduts who have successfully navigated societal challenges may be more likely to achieve economic stability and reside in advantaged areas [26]. Proximity to the academic medical center could play a role with individuals in low ADI areas potentially having easier access to care [27]. Employment patterns might also influence our findings, as there could be a higher proportion of university or medical center employees in our sample who may reside in low ADI areas and have better healthcare access [28]. Although not provided in our EHR data, educational attainment is another crucial factor to consider, as higher education levels are often associated with lower ADI scores and may influence healthcare-seeking behaviors and LGBTQ+ identity disclosure [29]. Lastly, better health insurance coverage among individuals living in lower ADI areas may facilitate access to care at academic medical centers [30]. Future research should examine these factors to develop a more nuanced understanding of the complexities between individual characteristics, socioeconomic status, and healthcare utilization among LGBTQ+ older adults, aligning with the SGM Health Disparities Framework.

**Interpersonal level factors.** The higher proportion of patients from low disadvantage areas who identified as lesbian or gay, compared to those identifying as asexual in moderate and high disadvantage areas, suggests potential differences in social support and community connection. This is supported by research showing that socioeconomic status (SES) significantly impacts social support networks among LGBTQ+ individual, with higher SES individuals from higher socioeconomic backgrounds being more likely to receive support from multiple sources, including family, peers, and significant others, while those from lower socioeconomic backgrounds often lack family support and experience worse mental health outcomes [31,32]. This aligns with the framework's emphasis on social networks and support as key determinants of health outcomes. Additionally, LGBTQ+ older adults from low socioeconomic disadvantaged areas may feel more safe, secure, and confident to disclose their sexual orientation and/or gender identity. Whereas, LGBTQ+ older adults who are from higher disadvantaged areas may not report their sexual orientation or gender identity for fear of receiving a lower standard of care or encountering discrimination [10].

**Community level factors.** Our findings highlight potential community barriers to healthcare access for LGBTQ+ older adults in more disadvantaged areas. The lower representation of patients from high disadvantage areas may indicate issues with healthcare accessibility or quality in these areas, rather than a lower need for care. Previous studies indicate that residents of moderate to high socioeconomic disadvantaged areas show more symptom burden of heart failure, but do not have increased hospitalizations compared to lower socioeconomic disadvantaged areas [9]. Transportation barriers in more deprived areas, including the cost of bus fare and travel time, may contribute to reduced access to healthcare [33].

**Societal level factors.** Although our study did not directly analyze societal factors, they play a crucial role in shaping health outcomes for LGBTQ+ older adults and are essential to consider when interpreting our results. Several key societal factors are relevant to our findings – *Policy and legal environment*: The unexpected distribution of patients across socioeconomic disadvantage areas may reflect broader policy influences. States and regions with more protective policies for LGBTQ+ individuals (e.g., anti-discrimination laws) may foster environments where LGBTQ+ older adults are more likely to achieve economic stability and access healthcare [2,34]. This could partially explain the higher representation of patients from low disadvantage areas in our sample. *Structural stigma*: The societal-level conditions, cultural norms, and institutional policies that constrain the opportunities and well-being of stigmatized populations may significantly influence our findings. LGBTQ+ older adults living in areas with high structural stigma may be less likely to disclose their sexual orientation or gender identity, potentially leading to underrepresentation in health data and reduced access to appropriate care. This could contribute to the lower representation of patients from high disadvantaged areas in our sample. *Healthcare system practices*: The lack of standardized, LGBTQ+ specific screening and care guidelines across healthcare systems may lead to inconsistent data collection and care provision. This could partly explain the high prevalence of individuals identifying as asexual in our study, potentially reflecting inadequate training in collecting sexual orientation or a lack of understanding among older adults about sexual orientation terminology. *Healthcare access and resource*

*allocation*: Societal decisions about healthcare resource allocation may contribute to disparities in access. As noted in our findings, LGBTQ+ older adults from more deprived areas may face reduced access to care due to transportation barriers and other socioeconomic factors. This aligns with experiences of other marginalized populations, such as Indigenous older adults, who face similar barriers to healthcare access [35].

These societal factors highlight the need for comprehensive policy approaches to address health disparities among LGBTQ+ older adults. They identify the importance of considering not just individual and community-level factors, but also broader societal influences when interpreting healthcare utilization patterns and developing interventions to improve health equity for this population.

## Future directions

Future research should focus on several key areas to address the complex interplay of factors affecting LGBTQ+ older adults' health outcomes. Researchers should examine how individual resilience factors interact with area-level disadvantage, potentially uncovering protective mechanisms that could inform interventions. Furthermore, understanding how older adults and healthcare workers define "asexual" would help us understand limitations in this study. Investigating barriers to hospital access for LGBTQ+ older adults in more disadvantaged areas is crucial to understanding and addressing healthcare disparities. Additionally, exploring the role of social support and community resources across different levels of area disadvantage could reveal important strategies for mitigating health inequities. Future studies should also assess the impact of structural factors, such as anti-discrimination laws and healthcare policies, on hospital utilization patterns for LGBTQ+ older adults in various socioeconomic contexts. While some studies might benefit from comparing LGBTQ+ and non-LGBTQ+ populations, it's equally important to conduct research that explores the rich diversity within LGBTQ+ older adult communities without necessarily using non-LGBTQ+ populations as a reference point. Longitudinal studies examining how health experiences and needs evolved over time could provide important information for developing long-term strategies to promote equity in aging populations. Finally, research should focus on patterns of care for specific health conditions and populations within the LGBTQ+ older adult community. For example, studying healthcare utilization and outcomes for transgender and non-binary older adults, as well as conditions like lung cancer, could provide valuable insights. These diverse research directions will contribute to a more comprehensive understanding of LGBTQ+ older adults' health needs and inform targeted interventions and policies.

## Limitations

A limitation of this study is that this patient population was from one hospital system and the results may not be generalizable to other LGBTQ+ older adult patients in other hospital systems. Additionally, we did not have access to a 9-digit zip code for all patient addresses; this prohibited the ability to calculate specific ADI codes at the street level. Street level data would have provided information on the socioeconomic disadvantage of the street the patient listed as their residence.

Secondary analysis of EHR data is a limitation of this study. EHRs are primarily designed for clinical care and billing purposes, not research, which can lead to data quality issues and potential bias [36]. The unexpected high prevalence of older adults identifying as asexual (60%) in our sample raises concerns about the accuracy of the sexual orientation data collection among older demographics at this academic medical center. This figure significantly differs from national and state-level data; for instance researchers reported that only 7.87% of older persons in North Carolina identify as asexual [37]. This discrepancy highlights the potential for misclassification or misunderstanding in SOGI data collection, a common issue in EHR-based research. These limitations underscore the need for improved SOGI data collection methods in healthcare settings [21], particularly for older adults, and caution in interpreting results from EHR-based studies.

Since the pandemic, there has been an increase in use of disadvantage indices, like ADI, but limited information on the differences in interpretation of these indices on the same geographical area. Depending on which disadvantage index is used, the social and policy response may vary [38]. In this study, there potentially could have been additional

disadvantage factors identified using a different index. Being that this is the first study to identify the socioeconomic disadvantage of LGBTQ+ older adults who use this health system, a future study should consider using more than one disadvantage index and comparing the two, looking at data from a longitudinal perspective and obtaining community feedback on results, and studying more closely geographical areas and circumstances within the area [38].

## Conclusion

In conclusion, while our findings challenge some assumptions about LGBTQ+ older adults and socioeconomic disadvantage, they also highlight the need for a nuanced understanding of health disparities within this population. By centering these results within the SGM Health Disparities Framework, we can better inform targeted interventions and policies to improve health equity for all LGBTQ+ older adults, regardless of their area's socioeconomic status. It's important to note that these findings have policy implications for resource allocation, emphasizing that areas of lower socioeconomic disadvantage should not necessarily receive more resources at the expense of more disadvantaged areas.

## Author contributions

**Conceptualization:** Jennifer T. May, Devon Noonan, Susan G. Silva.

**Data curation:** Jennifer T. May.

**Formal analysis:** Jennifer T. May, Susan G. Silva.

**Funding acquisition:** Jennifer T. May.

**Methodology:** Jennifer T. May, Susan G. Silva.

**Resources:** Jennifer T. May.

**Software:** Jennifer T. May.

**Supervision:** Susan G. Silva.

**Visualization:** Jennifer T. May, Susan G. Silva.

**Writing – original draft:** Jennifer T. May, Devon Noonan, Susan G. Silva.

**Writing – review & editing:** Jennifer T. May, Devon Noonan, Susan G. Silva.

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
