## [Decision Letter · Decision Letter 0]

12 Feb 2025

Dear Dr. May,

Thank you for submitting your manuscript to PLOS ONE. After careful consideration, we feel that it has merit but does not fully meet PLOS ONE’s publication criteria as it currently stands. Therefore, we invite you to submit a revised version of the manuscript that addresses the points raised during the review process.

<h3>**Hard Recommendations ** </h3>

**Justify the Exclusion of Emergency Room Visits**The Introduction references prior research on socioeconomic status and hospital/emergency room utilization. However, the Methods section does not clarify why emergency room visits were excluded. The authors must provide a strong rationale for this exclusion, as ER visits are a critical component of healthcare access, particularly in socioeconomically disadvantaged populations.**Clarify the Relationship Between ADI Groups and Primary Diagnoses/Hospitalizations per Patient**The Results section does not explicitly explore whether there is a correlation between ADI groups and specific health conditions or the frequency of hospitalizations per patient. If no relationships were found, this should be clearly stated. If such data exist but were not analyzed, the authors should consider including it, as it would enhance the study’s clinical relevance.**Provide References for Social Support and Community Connection Claims**The Discussion section states that differences in sexual orientation across ADI groups suggest differences in social support and community connection. However, no references are provided to support this claim. The authors must either cite relevant literature or provide empirical justification for these assertions.

<h3>**Soft Recommendations ** </h3>

**Improve Writing and Grammar Throughout the Manuscript**The manuscript contains multiple spelling and grammatical errors, as noted by Reviewer 2. Specific lines (99-101, 163-166, 192-193, etc.) require correction, but the entire manuscript should be reviewed for clarity and coherence. A professional language edit is strongly recommended.**Clarify the Placeholder Citation ("XX et al. 2023[21]")**On line 165, there is a reference to “(see XX et al. 2023[21]).” If this is an intentional blinding of author names for review purposes, it should be indicated. If it was left in by mistake, it must be corrected.**Discuss the Potential for Future Comparisons to Heterosexual and Cisgender Counterparts**While the Introduction makes comparisons between LGBTQ+ individuals and their heterosexual/cisgender counterparts, the Discussion does not return to this point. The authors should consider acknowledging this as a potential future research direction.

We look forward to receiving your revised manuscript.

Kind regards,

Harpreet Singh Grewal

Academic Editor

PLOS ONE

Journal Requirements:

2. Please update the “Ethics Statement” section in the Metadata of your manuscript with the relevant information."

“This work was supported by the National Center for Advancing Translational Sciences

(NCATS), National Institutes of Health, through Grant Award Number UL1 TR002553.

The content is solely the responsibility of the authors and does not necessarily

represent the official views of the NIH.”

“*This work was supported by the National Center for Advancing Translational Sciences (NCATS), National Institutes of Health, through Grant Award Number UL1 TR002553. The content is solely the responsibility of the authors and does not necessarily represent the official views of the NIH.**“*

“This work was supported by the National Center for Advancing Translational Sciences

(NCATS), National Institutes of Health, through Grant Award Number UL1 TR002553.

The content is solely the responsibility of the authors and does not necessarily

represent the official views of the NIH.”

6. We note that you have indicated that there are restrictions to data sharing for this study. For studies involving human research participant data or other sensitive data, we encourage authors to share de-identified or anonymized data. However, when data cannot be publicly shared for ethical reasons, we allow authors to make their data sets available upon request. For information on unacceptable data access restrictions, please see http://journals.plos.org/plosone/s/data-availability#loc-unacceptable-data-access-restrictions.

Reviewers' comments:

Reviewer's Responses to Questions

**Comments to the Author**

1. Is the manuscript technically sound, and do the data support the conclusions?

Reviewer #1: Yes

Reviewer #2: Yes

2. Has the statistical analysis been performed appropriately and rigorously?

Reviewer #1: Yes

Reviewer #2: I Don't Know

3. Have the authors made all data underlying the findings in their manuscript fully available?

Reviewer #1: Yes

Reviewer #2: No

4. Is the manuscript presented in an intelligible fashion and written in standard English?

Reviewer #1: Yes

Reviewer #2: No

Reviewer #1: This study conducted a secondary analysis of hospital admission records for which economic/neighborhood groups of LGBTQ+ older individuals were treated at some academic medical centers. Almost all patients were there for treatment of pulmonary symptoms. The majority of LGBTQ+ patents (66.4%) were from areas with low socioeconomic disadvantage. The authors expressed surprise about this finding. The authors discuss patient ease of access, support, insurance, and community factors facilitating their care saying policy should address LGBTQ+ patients from less affluent neighborhoods with higher social disadvantage. This is a reasonable recommendation. There is no discussion of health outcomes. Likely this information was not in the health records that they consulted. Is this a limitation?

Reviewer #2: This article aims to determine if a relationship exists between LGBTQ+ older adult hospitalizations and the county-level socioeconomic conditions in which they reside, and to interpret this relationship through the lens of the SGM Health Research Framework at the individual, interpersonal, community, and societal level. This reviewer believes the authors were mostly successful in accomplishing their aims. The manuscript appears technically sound and the data supports their conclusions; this reviewer is not well-versed enough in statistical analyses to address their appropriateness or rigor; the authors do not provide access to the data, citing unspecified privacy/ethical concerns; and while most of the manuscript is intelligible and written in standard English, there are a number of spelling/grammatical errors that require correction. The major and minor issues that should be addressed prior to publication are detailed below:

1. One major area of concern is the lack of justification for the exclusion of emergency room visits. In the Introduction, the authors reference previous studies linking socioeconomic status and rates of hospitalizations, rehospitalizations, and emergency visits. It is not made clear in the Methods section why emergency room visits were not included.

2. Relatedly, another major area of concern is in the authors’ examination of “ADI Groups and Patient Characteristics” in the Results section. The authors do not mention any analyses regarding relationships between ADI groups and primary diagnoses nor hospitalizations per patient. If no relationships were found, this should be stated.

3. Another area of concern: in the Discussion section, “Interpersonal Level Factors”, on lines 268-270, the authors do not adequately provide reasoning for why differences in sexual orientation across low, moderate, and high disadvantage areas suggests differences in social support and community connection. What is the reasoning behind this claim? No references are provided for differences in social support and community connection between lesbian/gay-identified older adults and asexual older adults.

4. Of more minor concern is the writing of this article. While much of the article is perfectly intelligible and well-written, there are multiple spelling and grammatical errors, and several sentences that are unclear throughout the paper. Please see the following lines for errors that should be addressed prior to publication: 99-101, 163-166, 192-193, 194, 201, 203, 228, 260, 280, 284, 290-292, 338-339, 346, 350, 352, 354. This reviewer also recommends the entire article be reviewed for any spelling mistakes and grammatical errors, in case any were missed in the above list.

5. Of minor concern: on line 165, the article says “(see XX et al. 2023[21])”. It is unclear if this is meant as a purposeful blinding, or if it was mistakenly left in.

6. Of minor concern: while comparisons to heterosexual and cisgender counterparts are made in the introduction section, the Discussion section does not bring this up as a potential future direction of research.

**Do you want your identity to be public for this peer review?** For information about this choice, including consent withdrawal, please see our Privacy Policy

Reviewer #1: No

Reviewer #2: No

---

## [Author Response · Author response to Decision Letter 1]

23 Apr 2025

Please find the reviewer’s comments and our responses below. Thank you for this opportunity to revise and resubmit.

1. Justify the Exclusion of Emergency Room Visits

The Introduction references prior research on socioeconomic status and hospital/emergency room utilization. However, the Methods section does not clarify why emergency room visits were excluded. The authors must provide a strong rationale for this exclusion, as ER visits are a critical component of healthcare access, particularly in socioeconomically disadvantaged populations.

Authors’ comments: Thank you for the comment. We agree that ED visits are a critical component of care and hope to look at this with future research studies. The focus of the study was on inpatient hospitalizations only. In the introduction the Armenia et al. (2017) article describes an observational study from the National Inpatient Sample database of the Healthcare Cost and Utilization Project (HCUP) data. The study discusses emergency surgeries of patients who are admitted as inpatient status. We have clarified throughout the manuscript by using the term “inpatient” before hospitalizations where appropriate, including the title.

2. Clarify the Relationship Between ADI Groups and Primary Diagnoses/Hospitalizations per Patient

The Results section does not explicitly explore whether there is a correlation between ADI groups and specific health conditions or the frequency of hospitalizations per patient. If no relationships were found, this should be clearly stated. If such data exist but were not analyzed, the authors should consider including it, as it would enhance the study’s clinical relevance.

Authors’ comments: We agree that exploring the relationship between ADI group and specific health conditions or frequency of hospitalization per patient would be useful. However, the study was not part of the information extracted from the electronic medical records. Thus, we did not conduct these analyses.

3. Provide References for Social Support and Community Connection Claims

The Discussion section states that differences in sexual orientation across ADI groups suggest differences in social support and community connection. However, no references are provided to support this claim. The authors must either cite relevant literature or provide empirical justification for these assertions.

Authors’ comments: We have added supporting information for this statement. The additional information reads: Pg 14, lines 267-272 - This is supported by research showing that socioeconomic status (SES) significantly impacts social support networks among LGBTQ+ individual, with higher SES individuals from higher socioeconomic backgrounds are more likely to receive support from multiple sources, including family, peers, and significant others, while those from lower socioeconomic backgrounds often lack family support and experience worse mental health outcomes.[31, 32]

Soft Recommendations

1. Improve Writing and Grammar Throughout the Manuscript

The manuscript contains multiple spelling and grammatical errors, as noted by Reviewer 2. Specific lines (99-101, 163-166, 192-193, etc.) require correction, but the entire manuscript should be reviewed for clarity and coherence. A professional language edit is strongly recommended.

Authors’ comments: Thank you for bringing these errors to our attention.

Line 57 – correction to the spelling of “identity”

Line 65 - correction to the spelling of “county”

Lines 97: We have added a comma after the word “model” in line 97 to now read as: “Guided by the social ecological model,[19]…”. This aligns with the Vancouver style of intext citations.

Line 100 the spelling of incapsulated has been corrected to “encapsulated”.

2. Clarify the Placeholder Citation ("XX et al. 2023[21]")

On line 165, there is a reference to “(see XX et al. 2023[21]).” If this is an intentional blinding of author names for review purposes, it should be indicated. If it was left in by mistake, it must be corrected.

Authors’ comments: Thank you. We have inserted “blinded for review” as a placeholder and have added this to the references list.

3. Discuss the Potential for Future Comparisons to Heterosexual and Cisgender Counterparts

While the Introduction makes comparisons between LGBTQ+ individuals and their heterosexual/cisgender counterparts, the Discussion does not return to this point. The authors should consider acknowledging this as a potential future research direction.

Authors’ comments: Thank you for this suggestion. Pg 17, lines 330-335 reads: While some studies might benefit from comparing LGBTQ+ and non-LGBTQ+ populations, it’s equally important to conduct research that explores the rich diversity within LGBTQ+ older adult communities without necessarily using non-LGBTQ+ populations as a reference point. Longitudinal studies examining how health experiences and needs evolved over time could provide important information for developing long-term strategies to promote equity in aging populations.

---

## [Decision Letter · Decision Letter 1]

8 Jul 2025

Dear Dr. May,

Thank you for submitting your manuscript to PLOS ONE. We invite you to submit a revised version of the manuscript that addresses the points raised during the review process.

We look forward to receiving your revised manuscript.

Kind regards,

Saima Aleem

Academic Editor

PLOS ONE

Journal Requirements:

Reviewers' comments:

Reviewer's Responses to Questions

**Comments to the Author**

Reviewer #1: All comments have been addressed

Reviewer #2: (No Response)

2. Is the manuscript technically sound, and do the data support the conclusions?

Reviewer #1: Yes

Reviewer #2: Yes

3. Has the statistical analysis been performed appropriately and rigorously?

Reviewer #1: Yes

Reviewer #2: Yes

4. Have the authors made all data underlying the findings in their manuscript fully available?

Reviewer #1: No

Reviewer #2: No

5. Is the manuscript presented in an intelligible fashion and written in standard English?

Reviewer #1: Yes

Reviewer #2: No

Reviewer #1: The review has addressed all of the reviewers' concerns and queries. The authors did a competent job in refining this submission.

Reviewer #2: A majority of the original comments were addressed. The manuscript is technically sound, the data support the conclusion, and the analyses were appropriate and rigorous. The authors indicate the data cannot be shared publicly because of data privacy, but is available upon request. However, minor revisions are recommended as there are still a large number of grammatical errors in the text that were not addressed. It is recommended the authors correct the following errors and review their manuscript carefully for any additional grammatical errors missed by the reviewer:

Line 165: “…or it can be entered it in the clinical setting…” to “…or can be entered in the clinical setting…”

Line 195-196: “…when there is a significant overall effect was detected.” to “…when a significant overall effect was detected.”

Line 243: “…the majority of (66.4%) of hospitalized LGBTQ+…” to “…the majority (66.4%) of hospitalized LGBTQ+…”

Line 276: “…backgrounds are more likely to receive support from multiple sources…” to “…backgrounds being more likely to receive support from multiple sources…”

Line 287: typo—change “my” to “may”

Line 291: typo—change “tavel” to “travel”

Line 298: typo—change “my” to “may”

Line 299: typo—change “aniti-discrimination” to “anti-discrimination”

Line 350-351: “Street level data would would have…” to “Street level data would have…”

Line 362: “…particularly or older…” to “…particularly for older…”

Line 364: “Since the pandemic there…” to “Since the pandemic, there…”

Line 366: “Depending on which disadvantage index used…” to “Depending on which disadvantage index is used…”

Except for these grammatical issues, the manuscript is well-written and engaging. In particular, the Discussion, Future Directions, and Limitations sections were well executed and robust.

**Do you want your identity to be public for this peer review?** For information about this choice, including consent withdrawal, please see our Privacy Policy

Reviewer #1: No

Reviewer #2: No

---

## [Author Response · Author response to Decision Letter 2]

28 Jul 2025

Reviewer Comments

Reviewer #1

1. The review has addressed all of the reviewers' concerns and queries. The authors did a competent job in refining this submission.

Thank you!

Reviewer #2

1. A majority of the original comments were addressed. The manuscript is technically sound, the data support the conclusion, and the analyses were appropriate and rigorous. The authors indicate the data cannot be shared publicly because of data privacy, but is available upon request. However, minor revisions are recommended as there are still a large number of grammatical errors in the text that were not addressed.

Thank you!

2. It is recommended the authors correct the following errors and review their manuscript carefully for any additional grammatical errors missed by the reviewer:

I appreciate the level of detail the errors are listed and the opportunity to correct these. I apologize for the oversight. We have carefully reviewed and provided an additional edit to correct the font of the reference page.

Line 165: “…or it can be entered it in the clinical setting…” to “…or can be entered in the clinical setting…”

Corrected.

Line 195-196: “…when there is a significant overall effect was detected.” to “…when a significant overall effect was detected.”

Corrected.

Line 243: “…the majority of (66.4%) of hospitalized LGBTQ+…” to “…the majority (66.4%) of hospitalized LGBTQ+…”

Corrected.

Line 276: “…backgrounds are more likely to receive support from multiple sources…” to “…backgrounds being more likely to receive support from multiple sources…”

Corrected.

Line 287: typo—change “my” to “may”

Corrected.

Line 291: typo—change “tavel” to “travel”

Corrected.

Line 298: typo—change “my” to “may”

Corrected.

Line 299: typo—change “aniti-discrimination” to “anti-discrimination”

Corrected.

Line 350-351: “Street level data would would have…” to “Street level data would have…”

Corrected.

Line 362: “…particularly or older…” to “…particularly for older…”

Corrected.

Line 364: “Since the pandemic there…” to “Since the pandemic, there…”

Corrected.

Line 366: “Depending on which disadvantage index used…” to “Depending on which disadvantage index is used…”

Corrected.

3. Except for these grammatical issues, the manuscript is well-written and engaging. In particular, the Discussion, Future Directions, and Limitations sections were well executed and robust.

Thank you!

---

## [Editor Report · Decision Letter 2]

5 Aug 2025

Social determinants of health in lesbian, gay, bisexual, transgender, queer, and other sexual and gender minority (LGBTQ+) older adults: Impact of socioeconomic disadvantage on inpatient hospitalizations

PONE-D-25-01218R2

Dear Dr. May,

We’re pleased to inform you that your manuscript has been judged scientifically suitable for publication and will be formally accepted for publication once it meets all outstanding technical requirements.

Kind regards,

Saima Aleem

Academic Editor

PLOS ONE
---

## [Editor Report · Acceptance letter]

PONE-D-25-01218R2

PLOS ONE

Dear Dr. May,

I'm pleased to inform you that your manuscript has been deemed suitable for publication in PLOS ONE. Congratulations! Your manuscript is now being handed over to our production team.

Kind regards,

on behalf of

Dr. Saima Aleem

Academic Editor

PLOS ONE